# Interaction of Clinical Factors Modestly Predict Anti-TNF-Alpha Antibody Formation in a Real-World Cohort of Inflammatory Bowel Disease Patients

**DOI:** 10.3390/biomedicines13112622

**Published:** 2025-10-26

**Authors:** Krisztián Kovács, Petra Nagypál, Barna Vásárhelyi, Antal Dezsőfi-Gottl, Nóra Béres, Pál Miheller, Ákos Iliás, Anna Balogh, Bence Prehoda, Luca Tóbi, Attila Szabó, Áron Cseh

**Affiliations:** 1Department of Laboratory Medicine, Semmelweis University, 1089 Budapest, Hungary; vasarhelyi.barna@semmelweis.hu; 2Paediatric Centre, Semmelweis University, 1083 Budapest, Hungary; nagypal.petra98@gmail.com (P.N.); dezsofi.antal@semmelweis.hu (A.D.-G.); beres.nora@semmelweis.hu (N.B.); annabalogh94@gmail.com (A.B.); prehoda.bence1@semmelweis.hu (B.P.); tobiluca02@gmail.com (L.T.); szabo.attila@semmelweis.hu (A.S.); cseh.aron@semmelweis.hu (Á.C.); 3Department of Surgery, Transplantation and Gastroenterology, Semmelweis University, 1082 Budapest, Hungary; miheller.pal@semmelweis.hu; 4Department of Internal Medicine and Oncology, Semmelweis University, 1088 Budapest, Hungary; ilias.akos@semmelweis.hu

**Keywords:** infliximab, adalimumab, immunogenicity, inflammatory bowel disease, Crohn’s disease, ulcerative Colitis, antidrug antibodies, ELISA

## Abstract

**Background:** Biological therapy is frequently used for the treatment of inflammatory bowel disease (IBD); however, the long-term efficacy of anti-tumor necrosis factor-alpha (TNF−α) therapies, such as infliximab (IFX) and adalimumab (ADA), is often compromised by the development of antidrug antibodies (AIFX and AADA, respectively). While several individual factors are known to contribute to immunogenicity, the complex, interactive effects of various clinical variables have not been fully elucidated in a real-world setting. **Methods:** We conducted a hierarchical logistic regression analysis on a retrospective cohort of 153 pediatric and adult IBD patients receiving IFX or ADA therapy to identify clinical factors and their interactions associated with AIFX/AADA positivity. The analysis progressively incorporated demographic, disease-related, and treatment-related variables, culminating in a model that included two- and three-way interaction terms. **Results:** Our final model demonstrated modest predictive power, with a Nagelkerke R^2^ of 0.287, explaining less than 30% of the variance in antibody positivity using readily available clinical data (AUC of 0.806, 71.0% sensitivity and 77.6% specificity). Key predictors included the type of biological therapy (IFX vs. ADA) and the duration of treatment, with IFX therapy being a significant independent predictor (OR = 6.940, *p* = 0.004) for antibody positivity. Importantly, we identified novel three-way interactions, revealing that the combined effect of age at disease onset, IBD subtype, and biological therapy type significantly influences antibody formation (*p* = 0.042), particularly in childhood-onset ulcerative colitis patients treated with IFX. A similar interaction was found for treatment duration, IBD subtype, and therapy type (*p* = 0.042), where the risk of antibody positivity with IFX increased significantly with treatment length, particularly in UC patients. **Conclusions:** This study highlights that the combination of routine clinical variables in IBD offers a data-driven, mechanistically insightful framework, supporting the prediction of AIFX/AADA positivity to a modest extent. This framework requires prospective and external validation before clinical implementation.

## 1. Introduction

Inflammatory bowel disease (IBD), comprising Crohn’s disease (CD) and ulcerative colitis (UC), is a group of chronic inflammatory conditions of the gastrointestinal tract. The pathophysiology of IBD is multifaceted and involves an abnormal immune response in the gut, leading to chronic inflammation. Tumor necrosis factor-alpha (TNF−α) plays a central role in the inflammatory cascade, and anti-TNF−α therapies, such as infliximab (IFX) and adalimumab (ADA), have revolutionized the management of IBD [1].

The effectiveness of these agents, however, is often limited by the development of antidrug antibodies (ADAs), which can reduce drug concentration and lead to secondary loss of response (sLOR) [2]. The presence of anti-infliximab antibodies (AIFX) is particularly concerning, as it has been linked to higher rates of treatment discontinuation and an increased risk of infusion-related adverse events [3,4]. While IFX is known to have a higher immunogenicity due to its chimeric structure, the formation of anti-adalimumab antibodies (AADAs) also remains a significant clinical concern [5,6].

The ability to identify patients at an increased risk of developing ADAs is crucial for optimizing treatment strategies and improving long-term outcomes. In a cohort study, patients who developed AIFX experienced a higher rate of treatment discontinuation due to loss of response compared to those without AIFX [7,8]. This report highlights the clinical importance of early detection of AIFX to make appropriate decisions on therapy modification.

ADA, which is another biologic against TNF-α, offers an alternative to IFX, but the presence of AADA has been correlated with decreased serum adalimumab concentrations and reduced clinical response rates. In the DIAMOND trial, a subanalysis revealed that patients with detectable AADA had significantly lower trough levels of ADA and diminished clinical remission rates [8]. Furthermore, a study demonstrated that higher ADA trough levels were associated with sustained clinical remission, whereas the presence of AADA predicted treatment failure [9]. This reinforces the utility of AADA detection in guiding therapeutic decisions.

There is a clear clinical need to acknowledge patient-specific factors that predispose to antibody development [10]. This would help the identification of patients who are at an increased risk of having AIFX and AADA. Despite the wealth of literature on individual risk factors, the complex interplay and synergistic effects between clinical variables have not been adequately explored in heterogeneous, real-world patient populations. Traditional multivariate methods often struggle to systematically assess the incremental value of adding high-order interaction terms informed by clinical theory. To address this gap, we hypothesized that the risk of immunogenicity is not merely an additive effect of known factors but is modulated by complex, synergistic high-order interactions among patient demographics, disease characteristics, and treatment regimens. We applied a hierarchical statistical modeling approach to systematically analyze the individual and interactive effects of various clinical parameters on the development of anti-TNF-alpha antibodies in a diverse cohort of pediatric and adult IBD patients.

## 2. Patients and Methods

### 2.1. Patients’ Enrollment

A cross-sectional study was conducted between June 2020 and December 2021, enrolling 40 children and 113 adults with IBD from the Paediatric Centre, Department of Surgery, Transplantation and Gastroenterology, and Department of Internal Medicine and Oncology at Semmelweis University. The inclusion criteria were a diagnosis of IBD based on the revised Porto criteria of the European Society for Paediatric Gastroenterology, Hepatology and Nutrition (ESPGHAN) for children and the diagnostic recommendations of the European Crohn’s and Colitis Organisation (ECCO) for adults, and were performed using a consecutive method [11,12]. The inclusion of both pediatric (n = 40) and adult (n = 113) patients reflects the spectrum of IBD care managed in a high-volume tertiary setting. Recognizing the critical difference in immunological development between these groups, the study design explicitly incorporated Age at Onset as a major predictor, specifically utilizing its interaction terms to model and account for potential age-related confounding effects on immunogenicity risk.

### 2.2. Ethical Statement

Data were collected for routine clinical care, and no additional blood samples were taken for the purpose of this study. Informed consent was obtained from all participants in this study, and all procedures were approved as declared in our ethical permission issued by the ethics committee (Medical Research Council, Ministry of Interior) with No. 19048-Á2/2018/EKU. The de-identification of data ensured the anonymity of the patients. All procedures performed in this study were in accordance with the 1964 Helsinki Declaration and its later amendments.

### 2.3. Methods

Before the receipt of the next IFX or ADA dose, 6 mL native blood samples were taken to measure trough IFX and ADA levels (TRL). Sera for IFX, ADA and antibody level detection were centrifuged (room temperature, 2300× *g*, 10 min). Samples were stored at −20 °C and used for ADA, AADA, IFX, and AIFX concentration measurements. For this purpose, LisaTracker Duo Infliximab and Duo Adalimumab In Vitro Diagnostic ELISA kits (Biosynex Theradiag, Croissy Beaubourg, France) were used. The assays were performed with the DAS APE ELITE ELISA instrument (DAS Instruments, Rome, Italy). The assays were performed according to the manufacturer’s specifications with internal quality controls run alongside patient samples to ensure procedural validity and reproducibility within the batch. Assay performance data are summarized in Table 1.

During induction therapy, target TRL were defined at least IFX 15 mg/L and at least ADA 7.5 µg/mL at 6 and 4 weeks, respectively. During maintenance therapy, target TRL ranges are 3–7 mg/L and 5–10 mg/L for IFX and ADA, respectively. For AIFX and AADA the cut-off levels are 9 µg/L and 4 µg/L, respectively, as proposed by ESPGHAN [14].

### 2.4. Statistical Analysis

A hierarchical logistic regression analysis was performed to identify clinical variables associated with anti-drug antibody positivity. The dependent variable was binary (Yes/No for AIFX/AADA presence). Continuous predictors (Age at Onset and Treatment Time) were standardized using a logarithmic transformation. Categorical predictors included Gender, IBD subtype (UC or CD), immunosuppressive therapy status, type of biological therapy (ADA or IFX), and dose intensification. This approach was utilized not for prediction optimization (as in stepwise regression) but to test a theory-driven, confirmatory hypothesis. Specifically, we aimed to determine the incremental variance (ΔR^2^) explained by treatment-related factors (M_3_) and high-order interactions (M_4_) over established, non-modifiable patient characteristics (M_1_ and M_2_). The progressive block design allows for a systematic assessment of whether the addition of complex interactions significantly improves model fit compared to simpler additive models, thereby confirming the hypothesized synergistic nature of immunogenicity risk.

The hierarchical model consisted of four steps: a reference model (M_0_), followed by M_1_ (adding Gender), M_2_ (adding Age at Onset, Localization, and IBD subtype), M_3_ (adding Treatment Time, immunosuppressive therapy, type of biological therapy, and dose intensification), and finally, M_4_, which incorporated two- and three-way interaction terms. Collinearity among predictors, including high-order interaction terms, was assessed using the Variance Inflation Factor (VIF). All VIF values for the predictors in the final model (M_4_) were below the critical threshold of 5.0 (Maximum VIF = 2.85), confirming acceptable model stability, see Table 2.

The analysis included several categorical variables (e.g., Localization, Immunosuppression status) containing notable proportions of missing data (“N/A”), as detailed in Table 3. To avoid listwise deletion, which would reduce the effective sample size and introduce selection bias, these “N/A” values were treated as a distinct, third categorical level within the respective predictor variables (e.g., Immunosuppression: Yes, No, N/A). We recognize that this approach may introduce limitations, which are discussed subsequently. Model comparisons were based on likelihood-ratio tests, and explanatory power was assessed using the Nagelkerke R^2^ value. The optimal cut-off value for the final model was determined using the Youden method to calculate sensitivity, specificity, and the area under the curve (AUC). All analyses were conducted in JASP (0.19.3) with a level of significance of 0.05.

## 3. Results

### 3.1. Patients’ Characteristics and Clinical Data

A total of 153 patients were included in the analysis, of whom 40 were children and 113 were adults. The clinical characteristics of the cohort are detailed in Table 3. Overall, 21% of patients (32/153) were positive for ADAs.

### 3.2. Model Fit and Hierarchical Structure

A hierarchical logistic regression was conducted to identify factors associated with AIFX/AADA positivity. The analysis comprised four models (M_0_–M_4_), each incorporating blocks of variables based on thematic relevance. For a detailed description of the performance of the models created in the block design, see Table 4.

The baseline model (M_0_) was completely empty and was used as a reference for comparing subsequent models. The first and second models (M_1_ and M_2_) introduced the Gender and Age at Onset, Localization, and IBD type, respectively, but these additions did not improve the model fitness.

The third model (M_3_), which incorporated treatment-related factors (Treatment Time, Immune suppression therapy, Type of Biological Therapy, and Intensification), showed a significant improvement in performance over the previous models (*p* = 0.002).

The final model (M_4_), which included interaction terms, further improved the model fit (*p* = 0.047). The Nagelkerke R^2^ value for M_4_ was 0.287, indicating that our selected clinical variables explain a partial 28.7% of the variance in antibody positivity. This indicates that a large portion of the variance (over 70%) remains unexplained by the clinical parameters utilized here. The model’s predictive power was confirmed by an AUC of 0.806, with a sensitivity of 71.0% and a specificity of 77.6% at the optimal cut-off value.

### 3.3. Significant Main Effects and Interactions

The final model identified several significant main effects and interaction terms, see Table 5. IFX therapy was a powerful predictor of antibody positivity, with patients receiving IFX being nearly seven times more likely to have detectable antibodies than those on ADA (OR = 6.940, *p* = 0.004). The duration of treatment was also a significant predictor, with a longer treatment period increasing the likelihood of antibody positivity (OR = 2.505, *p* = 0.029). Localization was also a significant factor (OR = 2.216, *p* = 0.049), suggesting that specific anatomical localizations are associated with AIFX/AADA antibody positivity.

In addition to the main effects, the final model included a number of interaction terms to explore possible synergistic effects between predictors, and four interactions reached statistical significance. The Treatment Time and Type of Biological Therapy interaction (OR = 0.220, *p* = 0.005) indicates that longer treatment appears to increase the risk of AADA, but not for AIFX. The interaction Age at Onset p × IBD type (OR = 8.023, *p* = 0.031) suggests that UC patients, variations in age at onset affect AIFX/AADA positivity, as AIFX/AADA positivity was more probable when IBD started in childhood.

The significance of three-way interaction, Age at Onset × IBD type × Type of Biological Therapy (OR = 0.019, *p* = 0.042), highlights the possibility of a complex relationship between age at onset, IBD type, and type of biologics used. This interaction demonstrates that the combined effect of childhood during onset and IBD type (UC or CD) on AIFX/AADA positivity is significantly influenced by the type of anti-TNF agent. However, it should be noted that the interpretation of the three-way interactions should be interpreted with caution due to the marginal *p* values (*p* = 0.042) and the limited sample size. For patients who were children at onset, IFX therapy was associated with a greater risk of AIFX/AADA positivity compared to ADA therapy. This effect is more significant in UC. To the contrary, in patients with adult-onset IBD, the difference in antibody positivity prevalence between ADA and IFX therapy diminishes in the UC group (Figure 1).

The significant three-way interaction between AIFX/AADA positivity and Treatment Time × IBD type × Type of Biological Therapy (OR = 82.745, *p* = 0.042) highlights the complex relationship between these variables. This suggests that longer treatment durations with IFX significantly predispose both CD and UC patients to a higher prevalence of antibody positivity compared to ADA therapy (Figure 2). Furthermore, it is essential to note that the three-way interaction involving Treatment Time exhibited a highly volatile odds ratio (OR = 82.745) with an extremely wide confidence interval (95% CI 1.250–5473.497). This wide interval strongly suggests limited precision and potential instability in modeling this highly specific subgroup, likely due to cell sparsity, and should be interpreted as purely exploratory rather than confirmatory.

## 4. Discussion

Our study reinforces the well-documented finding that IFX has a higher immunogenic potential compared to ADA. However, the primary value of our research lies in the identification of complex interactions between clinical factors that contribute to antibody formation in a real-world, heterogeneous patient population. Our hierarchical modeling approach allowed for a nuanced understanding of these relationships, which has not been widely explored in the existing literature. In our cross-sectional study, we analyzed and identified some factors that are associated with the presence of anti-TNF antibodies in UC and CD subjects. The hierarchical logistic regression model in the present study supports that treatment duration and type of biological therapy (IFX vs. ADA) are key predictors of anti-TNF antibody formation, which is consistent with previous studies published in the literature [15].

The finding of a significant three-way interaction involving age at onset, IBD subtype, and biological therapy is particularly notable. Previous research has suggested that IBD starting in childhood is associated with altered immune regulation, potentially leading to a heightened risk of anti-drug antibody formation [4,10]. Our findings expand on this by showing that this risk is further modulated by the specific anti-TNF agent used and the IBD subtype. The particularly high risk observed in childhood-onset UC patients on IFX aligns with the known higher immunogenicity of the chimeric IFX molecule compared to the humanized ADA molecule, suggesting that the naive, developing immune system in young UC patients may be highly reactive to the non-human epitopes present in IFX. This differential response underscores the need for personalized selection of biological agents in pediatric cohorts, particularly for UC, where IFX is often a frequent first-line choice [14].

The significant interaction between treatment duration and type of therapy also provides critical clinical insights. The increasing divergence in antibody prevalence between IFX and ADA with longer treatment duration underscores the importance of long-term Therapeutic Drug Monitoring (TDM) and antibody testing to preemptively address potential treatment failure [14]. This association is consistent with previous studies that demonstrate longer therapy periods enhance the immune response. These findings also support the importance of early detection of AIFX/AADA positivity and corresponding modification of therapy to prevent treatment failure in the long-term [4,5].

While our model achieved a statistically significant AUC of 0.806, the Nagelkerke R^2^ of 0.287 confirms that clinical factors alone provide only partial prediction. Dedicated prediction models incorporating critical pharmacokinetic variables—such as initial drug trough levels or genetic markers associated with drug metabolism—typically achieve higher predictive capability (e.g., AUCs exceeding 0.85). The key implication of our study, therefore, is not to replace these dedicated biomarker panels but to provide a readily accessible, clinical framework for early risk stratification, identifying patients who may benefit most from proactive TDM interventions [16,17,18,19]. The advantage of TDM has been proposed by many researchers, and according to the latest surveys, it is the pediatric IBD population in which is recommended and widely used for the optimization of therapy in the case of anti-TNF treatments [20,21].

The study’s design allowed us to enroll IBD patients of different ages and stages on ADA or IFX. Therefore, the results are characteristic of the population treated at a tertiary-care setting, where biological therapy treatment is provided for IBD, and they can be considered as “real-life” observations.

## 5. Study Limitations

This study is subject to several limitations inherent to its retrospective and cross-sectional design. First, the observed associations between clinical factors and antibody presence do not establish causality; thus, we cannot infer that longer treatment duration causes antibody formation, only that longer duration is predicted by or associated with antibody presence at the time of sampling. Prospective validation is essential before any causal inference can be made or clinical guidelines derived. Second, despite recruiting from a tertiary center, the overall cohort size (N = 153) remains modest, especially when exploring high-order, three-way interaction terms. This limited power manifests in the extremely wide confidence intervals observed for some interaction effects (Table 5), indicating instability and limited precision in modeling these specific, sparse subgroups. The results derived from these complex interactions must therefore be treated as strictly hypothesis-generating, requiring confirmation in larger, well-powered, and externally validated cohorts. Third, the reliance on retrospective clinical data introduced notable proportions of missing data (“N/A”) for several variables (e.g., 5-ASA status, Immunosuppression status). Although these were handled by treating “N/A” as a distinct category, this approach risks introducing bias associated with missingness, which may further contribute to the unexplained variance (71.3%) in the model.

## 6. Conclusions

In conclusion, our study confirms that anti-TNF-alpha antibody formation is a major challenge in IBD management, with IFX therapy and treatment duration being key predictors. The most significant finding, however, is the discovery of complex, synergistic interactions between easily accessible clinical factors, which provide a significant, albeit modest, predictive value. Our hierarchical modeling approach offers a framework for integrating multiple patient-specific variables to better assess the risk of immunogenicity. This framework requires prospective and external validation before it can be effectively integrated into routine clinical support systems for optimizing patient monitoring and preventing therapeutic failure in IBD care.

## Figures and Tables

**Figure 1 biomedicines-13-02622-f001:**
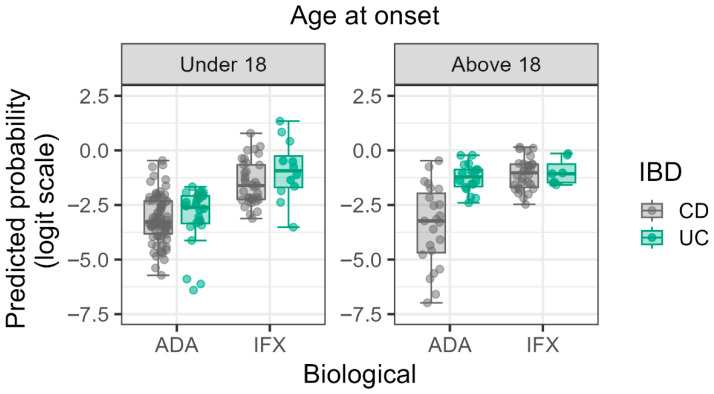
Logit predicted probabilities of antibody positivity stratified by age at onset, biological therapy, and IBD type. Predicted log probabilities are plotted on the *y*-axis, with biological therapies (ADA and IFX) on the *x*-axis. Gray dots represent individual Crohn’s disease (CD) patients and green dots represent individual Ulcerative Colitis (UC) patients. The plot illustrates the complex three-way interaction, particularly the increased risk associated with IFX in childhood-onset UC.

**Figure 2 biomedicines-13-02622-f002:**
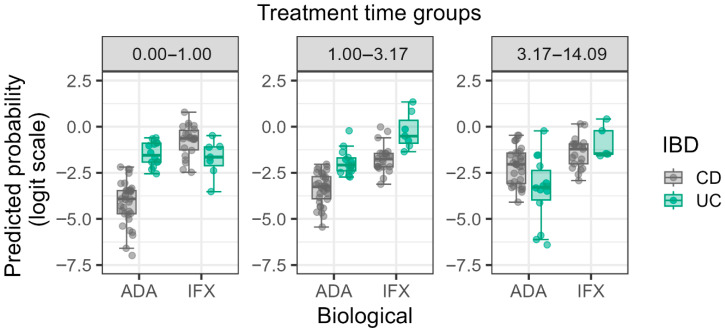
Logit predicted probabilities of antibody positivity stratified by treatment time, anti-TNF-alpha therapy, and IBD status. The panels represent three treatment time quartiles derived from the log-transformed data: Q1 (0.00–1.00 years), Q2 (1.00–3.17 years), and Q3 (3.17–14.09 years), respectively. Predicted log probabilities are plotted on the *y*-axis, with biological therapies (ADA and IFX) on the *x*-axis. Gray dots represent individual Crohn’s disease (CD) patients and green dots represent individual Ulcerative Colitis (UC) patients.

**Table 1 biomedicines-13-02622-t001:** Limit of detection (LoD) and assay range data of duo adalimumab and duo infliximab ELISA assays [13].

Limit of detection (LoD)	Adalimumab/Infliximab 0.3 mg/L (>95th percentile)	anti-Adalimumab/anti-Infliximab 10 µg/L (>95th percentile)
Assay range	Adalimumab/Infliximab 0.3 mg/L–20 mg/L	anti-Adalimumab/anti-Infliximab 10 µg/L–160 µg/L/10 µg/L–200 µg/L

**Table 2 biomedicines-13-02622-t002:** Model Diagnostic Metrics for the Final Logistic Regression Model (M_4_).

Diagnostic Test	Value	Interpretation
Hosmer–Lemeshow Goodness-of-Fit Test (*p*-value)	0.518	Confirms adequate model calibration (*p* > 0.05).
Maximum VIF (Variance Inflation Factor)	2.85	Indicates minimal multicollinearity.
Model Performance (Nagelkerke R^2^)	0.287	Indicates partial variance explanation.

**Table 3 biomedicines-13-02622-t003:** Patients’ characteristics. Clinical data normalized to the number of patients (excluding multiple measurement points from one patient): IFX vs. ADA in CD vs. UC sample. Abbreviations: 5-ASA: 5-aminosalicylates; ADA: adalimumab; CD: Crohn’s disease; IBD: inflammatory bowel disease; IFX: infliximab; UC: ulcerative colitis; Localization (CD): L1: Terminal ileum; L2: Colon; L3: Ileocolon; L4: Upper gastrointestinal (modifier for L1; L2; or L3); (UC): E1: Ulcerative proctitis (rectum only; E2: Left-sided colitis (up to splenic flexure); E3: Extensive colitis (beyond splenic flexure). Attitude: B1: Non-stricturing; non-penetrating; B2: Stricturing; B3: Penetrating. Severity: S0: Clinical remission; S1: Mild UC (≤4 bloody stools daily; no systemic toxicity; S2: Moderate UC (>4–5 stools daily; minimal systemic toxicity).

	CD	UC	Total
IFX	ADA	IFX	ADA	
**Number of patients (m/f)**	50 (27/23)	59 (23/36)	13 (6/7)	31 (14/17)	153 (70/83)

** Children**	10 (6%)	18 (12%)	5 (3%)	7 (5%)	40 (26%)
**Adult**	40 (26%)	41 (27%)	8 (5%)	24 (16%)	113 (74%)
**Childhood onset**	24 (16%)	37 (24%)	5 (3%)	10 (7%)	76 (50%)
**Disease initial age (year)**	19.1 [11.3–25]	15.3 [12.5–21.8]	22.5 [15.7–32.9]	24.1 [14.8–40.8]	18.6 [12.2–27.7]
**Time from the onset of the disease to biological therapy (year)**	5.09 [3.6–12.9]	3.92 [1.4–11.5]	4.16 [1.2–8.9]	3.59 [1.5–10.4]	4.42 [1.7–11.3]
**Antibody against IFX or ADA:**					
Yes	16 (11%)	5 (3%)	5 (3%)	6 (4%)	32 (21%)
No	34 (22%)	54 (36%)	8 (5%)	25 (16%)	121 (79%)
**Age at sampling (year)**	31.9 [23.4–41]	26.6 [17.8–38.8]	32.5 [17.2–45.1]	33.98	30.7 [18–41.7]
[18.6–53]
**Localization:**					
L1/E1	7 (4%)	11 (7%)	1 (1%)	-	19 (12%)
L2/E2	16 (10%)	14 (9%)	6(4%)	12 (8%)	48 (31%)
L3–4/E3–4	21 (14%)	29 (19%)	5 (3%)	18 (12%)	73 (48%)
N/A	6 (4%)	5 (3%)	1 (1%)	1 (1%)	13 (9%)
**Attitude:**					
B1/S0	26 (17%)	25 (16%)	5 (3%)	14 (9%)	68 (45%)
B2/B3/S1	21 (14%)	30 (19%)	7 (5%)	17 (11%)	75 (49%)
N/A	3 (2%)	4 (3%)	1 (1%)	-	10 (6%)
**Immunosuppression:**					
Yes	21 (14%)	33 (22%)	6 (4%)	13 (8%)	73 (48%)
No	16 (10%)	20 (13%)	4 (3%)	11 (7%)	51 (33%)
N/A	13 (8%)	6 (4%)	3 (2%)	7 (5%)	29 (19%)
**Steroid:**					
Yes	9 (6%)	8 (5%)	3 (2%)	6 (4%)	26 (17%)
No	27 (17%)	44 (29%)	7 (5%)	18 (12%)	96 (63%)
N/A	14 (9%)	7 (5%)	3 (1%)	7 (5%)	31 (20%)
**5-ASA:**					
Yes	9 (6%)	21 (14%)	7 (5%)	18 (12%)	55 (36%)
No	7 (5%)	20 (13%)	2 (1%)	3 (2%)	32 (21%)
N/A	34 (22%)	18 (12%)	4 (3%)	10 (6%)	66 (43%)
**Intensification:**					
Yes	11 (7%)	13 (8%)	4 (3%)	8 (5%)	36 (23%)
No	35 (23%)	44 (29%)	7 (5%)	22 (14%)	108 (71%)
N/A	4 (3%)	2 (1%)	2 (1%)	1 (1%)	9 (6%)

**Table 4 biomedicines-13-02622-t004:** Hierarchical Model Structure. The table shows the *p*-value and the efficiency of the fit of the hierarchically constructed models (M_0_–M_4_) via Nagelkerke R^2^ values. The *p* values in the table indicate whether the model’s performance significantly improved in comparison with the previous model.

Model	*p*	R^2^	Variables
M0			
M_1_	0.701	0.001	Gender
M_2_	0.132	0.025	M1 + Age at onset, Localization, IBD type
M_3_	0.002	0.107	M2 + Treatment Time, Immune suppression therapy, Type of Biological Therapy, Intensification
M_4_	0.043	0.291	M3 + Interaction terms

**Table 5 biomedicines-13-02622-t005:** Significant factors in the final logistic regression model (M_4_).

Variable	Odds Ratio (OR)	95% Confidence Interval	*p*-Value
Type of Biological Therapy (IFX vs. ADA)	6.940	1.890–25.467	0.004
Treatment Time	2.505	1.107–5.672	0.029
Localization	2.216	1.002–4.902	0.049
Interaction: Treatment Time × Type of Biological Therapy	0.220	0.076–0.638	0.005
Interaction: Age at Onset × IBD Type	8.023	1.222–52.684	0.031
Interaction: Age at Onset × IBD Type × Type of Biological Therapy	0.019	0.001–0.725	0.042
Interaction: Treatment Time × IBD Type × Type of Biological Therapy	82.745	1.250–5473.497	0.042

## Data Availability

All data analyzed during this study are included in this article.

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
