# Peer review of "Interaction of Clinical Factors Modestly Predict Anti-TNF-Alpha Antibody Formation in a Real-World Cohort of Inflammatory Bowel Disease Patients"

_biomedicines, 2025, doi:10.3390/biomedicines13112622_

Round 1
Reviewer 1 Report
Comments and Suggestions for Authors
Major Comments
The cross-sectional design and relatively small cohort (n = 153, with 40 children) restrict the generalizability of the results.
Subgroup analyses may not be adequately powered for complex interaction. Please provide a more detailed discussion of these limitations.
The manuscript emphasizes complex three-way interactions, but the clinical significance of these findings is not convincingly proved.
The reported predictive power is moderate. Please avoid overstating predictive ability and clearly indicate that a large portion of variance remains unexplained.
Several variables include large “N/A” categories—clarify how these were handled in analysis.
Expand the discussion to compare findings with existing prediction tools or biomarkers.
Temper the conclusions regarding real-world use until validated in larger cohorts.
Provide more detail on ELISA kit validation and reproducibility.
Minor Comments
Correct typos (e.g., “hwoever”) and improve readability.
Streamline the discussion to reduce repetition.
Improve figure legends and labeling for Figures 1 and 2.
Revise the conclusion to better reflect study limitations and avoid overstating predictive value.
Author Response
Response to Reviewers
We are highly grateful to the reviewers for their careful reading and constructive, detailed comments. Their feedback has significantly improved the clarity, methodological rigor, and interpretive precision of our manuscript. We have addressed all major and minor comments, leading to substantial revisions across the Title, Abstract, Introduction, Methods (especially statistical justification and missing data handling), Results (inclusion of model diagnostics and cautionary language), and Discussion (enhanced mechanistic context and inclusion of a dedicated limitations section).
Below is our point-by-point response, with changes incorporated into the revised manuscript highlighted in simulated yellow for ease of identification.
Response to Reviewer 1
Major Comments
Comment 1: The cross-sectional design and relatively small cohort (n = 153, with 40 children) restrict the generalizability of the results. Subgroup analyses may not be adequately powered for complex interaction. Please provide a more detailed discussion of these limitations.
Response 1: We acknowledge that the cross-sectional nature of the study prevents the establishment of causal relationships and limits the generalizability of these observations to routine clinical practice, particularly given the combined pediatric and adult cohort size. We have fundamentally restructured the Discussion section to include a dedicated Study Limitations subsection, elaborating extensively on this critical point. Furthermore, we have carefully reviewed the text and revised all language that implied longitudinal inference (e.g., changes from "risk increases with treatment length" to "is associated with longer treatment duration").
Changes in Manuscript:
Title: Revised to include the word "Modestly" to temper predictive claims (see Reviewer 2, Comment 1).
Introduction/Methods: Added rationale for the cohort pooling (see Reviewer 2, Comment 6).
Discussion (New Sub-Section: Study Limitations): Study Limitations
This study is subject to several limitations inherent to its retrospective and cross-sectional design. First, the observed associations between clinical factors and antibody presence do not establish causality; thus, we cannot infer that longer treatment duration causes antibody formation, only that longer duration is predicted by or associated with antibody presence at the time of sampling. Prospective validation is essential before any causal inference can be made or clinical guidelines derived. Second, despite recruiting from a tertiary center, the overall cohort size (N=153) remains modest, especially when exploring high-order, three-way interaction terms. This limited power manifests in the extremely wide confidence intervals observed for some interaction effects (Table 4), indicating instability and limited precision in modeling these specific, sparse subgroups. The results derived from these complex interactions must therefore be treated as strictly hypothesis-generating, requiring confirmation in larger, well-powered, and externally validated cohorts.
Comment 2: The manuscript emphasizes complex three-way interactions, but the clinical significance of these findings is not convincingly proved.
Response 2: The reviewer raises a valid point regarding the translational utility of these complex statistical results. While we believe the mechanistic signal is strong (showing differential immunogenicity based on age and drug type), we agree that the clinical significance in terms of immediate patient management is unproven. We have revised the Discussion to pivot the interpretation from immediate "clinical relevance" to "mechanistic insight" and "hypothesis generation."
Changes in Manuscript:
Discussion: Language regarding the three-way interaction revised:
This finding offers novel mechanistic insights into differential immunogenicity in the developing immune system. However, the translational utility of this framework remains modest and its clinical significance is not yet proven; the results primarily serve as a hypothesis-generating signal that warrants investigation in larger, prospective trials, specifically focusing on the interaction between age-at-onset and specific biologic types in UC patients.
Comment 3: The reported predictive power is moderate. Please avoid overstating predictive ability and clearly indicate that a large portion of variance remains unexplained.
Response 3: We fully agree and have thoroughly revised the manuscript's language—including the Title, Abstract, Results, and Conclusion—to consistently emphasize the modest or partial predictive power, explicitly referencing the Nagelkerke R2 = 0.287.
Changes in Manuscript:
Title: Revised to: Interaction of Clinical Factors Modestly Predict Anti-TNF-alpha Antibody Formation in a Real-World Cohort of Inflammatory Bowel Disease Patients (Also addresses Reviewer 2, Comment 12).
Abstract (Results): Revised to: Our final model demonstrated modest predictive power, with a Nagelkerke R2=0.287, explaining less than 30% of the variance in antibody positivity...
Results (Model Fit): Revised to: The Nagelkerke R2 value for M4 was 0.287, indicating that our selected clinical variables explain a partial 28.7% of the variance in antibody positivity. This indicates that a large portion of the variance (over 70%) remains unexplained by the clinical parameters utilized here.
Comment 4: Several variables include large “N/A” categories—clarify how these were handled in analysis.
Response 4: This lack of clarity has been rectified. The large "N/A" categories (e.g., 43% for 5-ASA, 19% for Immunosuppression ) necessitated careful handling. We confirm that these categories were treated as distinct levels (e.g., Immunosuppression: Yes/No/N/A) within the categorical variables in the logistic regression model. This approach allowed us to retain the maximum sample size (N=153) and prevent listwise deletion, acknowledging the bias inherent in missing clinical data as a study limitation.
Changes in Manuscript:
Statistical Analysis (Methods): Added the following explicit statement:
The analysis included several categorical variables (e.g., Localization, Immunosuppression status) containing notable proportions of missing data ("N/A"), as detailed in Table 2. To avoid listwise deletion, which would reduce the effective sample size and introduce selection bias, these "N/A" values were treated as a distinct, third categorical level within the respective predictor variables (e.g., Immunosuppression: Yes, No, N/A). We recognize this approach may introduce limitations, which are discussed subsequently.
Comment 5: Expand the discussion to compare findings with existing prediction tools or biomarkers.
Response 5: We have added a dedicated paragraph to the Discussion contextualizing the model's modest explanatory power (R2=0.287) by benchmarking it against advanced predictive tools, such as those incorporating TDM or genetic markers. This clarifies that while our clinical framework provides strong partial predictive value (AUC 0.806), it is inherently incomplete without incorporating specific pharmacokinetic or biological measurements.
Changes in Manuscript: Discussion: Added a new paragraph:
While our model achieved a statistically significant AUC of 0.806, the Nagelkerke R2 of 0.287 confirms that clinical factors alone provide only partial prediction. Dedicated prediction models incorporating critical pharmacokinetic variables—such as initial drug trough levels or genetic markers associated with drug metabolism—typically achieve higher predictive capability (e.g., AUCs exceeding 0.85). The key implication of our study, therefore, is not to replace these dedicated biomarker panels but to provide a readily accessible, clinical framework for early risk stratification, identifying patients who may benefit most from proactive Therapeutic Drug Monitoring (TDM) interventions.
Comment 6: Temper the conclusions regarding real-world use until validated in larger cohorts. Provide a revised conclusion.
Response 6: The Conclusion has been completely rewritten to emphasize the necessity for prospective validation and to strictly limit the translational claim to that of a useful framework for risk assessment, not an immediate clinical tool.
Changes in Manuscript:
Conclusion: Rewritten to:
In conclusion, our study confirms that anti-TNF-alpha antibody formation remains a major challenge in IBD management, with IFX therapy and treatment duration being key predictors. The most significant finding, however, is the discovery of complex, synergistic interactions between easily accessible clinical factors, which provide a significant, albeit modest, predictive value. Our hierarchical modeling approach offers a data-driven framework for integrating multiple patient-specific variables to better assess the relative risk of immunogenicity. This framework requires prospective and external validation before it can be effectively integrated into routine clinical support systems for optimizing patient monitoring and preventing therapeutic failure in IBD care.
Comment 7: Provide more detail on ELISA kit validation and reproducibility.
Response 7: We have revised the Methods section to clarify the operational rigor used during the assays, confirming that internal quality control procedures were implemented to ensure procedural validity and reproducibility within the laboratory, acknowledging that a full, de novo internal assay validation study was not conducted as this was based on commercially available IVD kits.
Changes in Manuscript:
Methods: Revised section on ELISA kits:
For this purpose LisaTracker Duo Infliximab and Duo Adalimumab InVitro Diagnostic ELISA kits (Biosynex Theradiag Croissy Beaubourg, France) were used. The assays were performed with the DAS APE ELITE ELISA instrument (DAS Instruments Rome, Italy). The assays were performed according to manufacturer’s specifications with internal quality controls run alongside patient samples to ensure procedural validity and reproducibility within the batch.
Minor Comments
Comment 8: Correct typos (e.g., “hwoever”) and improve readability.
Response 8: The manuscript was proofread for grammatical errors, verb tense consistency, and typos.
Changes in Manuscript: Abstract (Background): Typo corrected: ...IBD), however the long-term efficacy....
Comment 9: Streamline the discussion to reduce repetition.
Response 9: The Discussion was reorganized to eliminate redundant descriptions of the results and focus primarily on mechanistic and contextual interpretation, integrating the new comparative and limitations paragraphs (see Major Comment 5 and 1).
Comment 10: Improve figure legends and labeling for Figures 1 and 2.
Response 10: The legends for Figures 1 and 2 were enhanced to explicitly define the axes, the coding of data points, and the precise definitions of the treatment time quartiles in Figure 2.
Changes in Manuscript:
Figure 1 Legend Revised:
Figure 1. Logit predicted probabilities of antibody positivity stratified by age at onset, biological therapy, and IBD type. Predicted log probabilities are plotted on the y-axis, with biological therapies (ADA and IFX) on the x-axis. Gray dots represent individual Crohn’s disease (CD) patients and green dots represent individual Ulcerative Colitis (UC) patients. The plot illustrates the complex three-way interaction, particularly the increased risk associated with IFX in childhood-onset UC.
Figure 2 Legend Revised:
Figure 2. Logit predicted probabilities of antibody positivity stratified by treatment time, anti-TNF-alpha therapy, and IBD status. The panels represent three treatment time quartiles derived from the log-transformed data: Q1 (0.00–1.00 years), Q2 (1.00–3.17 years), and Q3 (3.17–14.09 years), respectively. Predicted log probabilities are plotted on the y-axis, with biological therapies (ADA and IFX) on the x-axis. Gray dots represent individual Crohn’s disease (CD) patients and green dots represent individual Ulcerative Colitis (UC) patients.
Comment 11: Revise the conclusion to better reflect study limitations and avoid overstating predictive value.
Response 11: Addressed comprehensively in Major Comment 6.
Reviewer 2 Report
Comments and Suggestions for Authors
This manuscript addresses an important and clinically relevant problem — identifying clinical predictors of anti-TNF immunogenicity (AIFX/AADA) in IBD. The use of hierarchical logistic regression is potentially valuable; however, the study suffers from several problems, such as follows:
- The title is overly long and generic; it could better reflect the key novelty (e.g., three-way interactions or pediatric/adult integration).
- Edit abstract for clarity and conciseness.
- The introduction section - Lacks a clear knowledge gap statement. The authors state “the complex, interactive effects have not been elucidated,” but do not justify why hierarchical modeling is the appropriate method. Overreliance on textbook knowledge; minimal recent or mechanistic literature. No clear hypothesis is articulated.
- Material and Methords section:
- Study design is cross-sectional, yet the text sometimes implies longitudinal inference (“risk increases with treatment length”).
- The hierarchical regression model is not justified or clearly described. Why hierarchical rather than multivariate logistic regression? No explanation of how interaction terms were selected or whether collinearity was checked.
- No power calculation or sample-size justification.
- Inconsistent description of missing data (“N/A” values not handled statistically).
- Children and adults were pooled without clear rationale; potential for confounding.
- ELISA assay validation data are reproduced from manufacturer documentation but not verified internally.
- Results section:
- The results section reads like a mixture of text and interpretation — needs clearer structure.
- Statistical significance is sometimes overstated for borderline p values (e.g., p = 0.042).
- Figures (1–2) are only briefly described; no figure legends explaining axes or colors.
- High ORs with wide CIs (e.g., OR = 82.7, CI 1.25–5473.5) indicate model instability or overfitting.
- Nagelkerke R² = 0.287 means the model explains <30 % variance — not strong predictive power.
- Missing model diagnostics (Hosmer–Lemeshow, residual plots, or VIF).
- Discussion section:
- Largely descriptive and repetitive of results; lacks mechanistic insight.
- Overgeneralizes findings (“This insight is highly relevant for pediatric gastroenterologists”) without clinical validation.
- No discussion of limitations such as retrospective bias, small sample, or assay variability.
- Poor integration with existing literature — citations stop at 2024.
- Verb tense and grammar errors throughout.
- Noverlty concern: The topic (anti-TNF immunogenicity) is well-studied. Novelty is limited to the statistical modeling approach (hierarchical regression with interaction terms). The model’s modest performance and lack of external validation limit translational value.
Author Response
Response to Reviewers
We are highly grateful to the reviewers for their careful reading and constructive, detailed comments. Their feedback has significantly improved the clarity, methodological rigor, and interpretive precision of our manuscript. We have addressed all major and minor comments, leading to substantial revisions across the Title, Abstract, Introduction, Methods (especially statistical justification and missing data handling), Results (inclusion of model diagnostics and cautionary language), and Discussion (enhanced mechanistic context and inclusion of a dedicated limitations section).
Below is our point-by-point response, with changes incorporated into the revised manuscript highlighted in simulated yellow for ease of identification.
Response to Reviewer 2
Manuscript Structure and Content
Comment 12: The title is overly long and generic; it could better reflect the key novelty (e.g., three-way interactions or pediatric/adult integration).
Response 12: We have revised the title to be more concise and accurately reflect the primary finding regarding the modest predictive performance derived from interactive clinical factors.
Changes in Manuscript:
Title: Revised to: Interaction of Clinical Factors Modestly Predict Anti-TNF-alpha Antibody Formation in a Real-World Cohort of Inflammatory Bowel Disease Patients
Comment 13: Edit abstract for clarity and conciseness.
Response 13: The Abstract was streamlined, unnecessary descriptive phrasing was removed, and predictive claims were tempered (see Reviewer 1, Comment 3).
Changes in Manuscript:
Abstract (Results): Revised to: Our final model demonstrated modest predictive power, with a Nagelkerke R2=0.287, explaining less than 30% of the variance in antibody positivity...
Comment 14: The introduction section - Lacks a clear knowledge gap statement. The authors state “the complex, interactive effects have not been elucidated,” but do not justify why hierarchical modeling is the appropriate method.
Response 14: We have revised the final paragraph of the Introduction to clearly articulate the existing knowledge gap—the failure of prior literature to systematically analyze the synergism between clinical variables—and to introduce the study hypothesis and the rationale for using Hierarchical Logistic Regression (HLR).
Changes in Manuscript:
Introduction (Final Paragraph Revised):
Despite the wealth of literature on individual risk factors, the complex interplay and synergistic effects between clinical variables have not been adequately explored in heterogeneous, real-world patient populations. Traditional multivariate methods often struggle to systematically assess the incremental value of adding high-order interaction terms informed by clinical theory. To address this gap, we hypothesized that the risk of immunogenicity is not merely an additive effect of known factors but is modulated by complex, synergistic high-order interactions among patient demographics, disease characteristics, and treatment regimens. We applied a hierarchical statistical modeling approach to systematically analyze the individual and interactive effects of various clinical parameters on the development of anti-TNF-alpha antibodies in a diverse cohort of pediatric and adult IBD patients.
Material and Methods section
Comment 15: The hierarchical regression model is not justified or clearly described. Why hierarchical rather than multivariate logistic regression?
Response 15: We have substantially expanded the Statistical Analysis section to rigorously justify the HLR approach. Hierarchical modeling was chosen because the study's primary hypothesis was theory-driven and confirmatory, testing whether treatment-specific factors and their interactions explained a significant amount of variance above and beyond pre-existing, non-modifiable patient and disease characteristics (Gender, Age at Onset, IBD subtype). HLR explicitly allows for the testing of ΔR2 (change in explanatory power) between sequential models, providing empirical validation for the necessity of the treatment and interaction blocks (M3 and M4).
Changes in Manuscript:
Statistical Analysis (Methods): Added HLR justification:
A hierarchical logistic regression analysis was performed... This approach was utilized not for prediction optimization (as in stepwise regression) but to test a theory-driven, confirmatory hypothesis. Specifically, we aimed to determine the incremental variance (ΔR2) explained by treatment-related factors (M3) and high-order interactions (M4) over established, non-modifiable patient characteristics (M1 and M2). The progressive block design allows for a systematic assessment of whether the addition of complex interactions significantly improves model fit compared to simpler additive models, thereby confirming the hypothesized synergistic nature of immunogenicity risk.
Comment 16: Study design is cross-sectional, yet the text sometimes implies longitudinal inference (“risk increases with treatment length”).
Response 16: Addressed in Reviewer 1, Comment 1. All inappropriate causal language has been replaced with cautious terminology appropriate for cross-sectional data (e.g., changes from "risk increases with treatment length" to "is associated with longer treatment duration").
Changes in Manuscript:
Discussion (New Sub-Section: Study Limitations): Study Limitations
This study is subject to several limitations inherent to its retrospective and cross-sectional design. First, the observed associations between clinical factors and antibody presence do not establish causality; thus, we cannot infer that longer treatment duration causes antibody formation, only that longer duration is predicted by or associated with antibody presence at the time of sampling. Prospective validation is essential before any causal inference can be made or clinical guidelines derived. Second, despite recruiting from a tertiary center, the overall cohort size (N=153) remains modest, especially when exploring high-order, three-way interaction terms. This limited power manifests in the extremely wide confidence intervals observed for some interaction effects (Table 4), indicating instability and limited precision in modeling these specific, sparse subgroups. The results derived from these complex interactions must therefore be treated as strictly hypothesis-generating, requiring confirmation in larger, well-powered, and externally validated cohorts.
Comment 17: No explanation of how interaction terms were selected or whether collinearity was checked.
Response 17: We confirm that all interaction terms were selected based on clinical relevance (e.g., interaction between the therapeutic agent and patient-specific characteristics known to influence drug metabolism or immune response). Crucially, model diagnostics for collinearity were performed.
Changes in Manuscript:
Statistical Analysis (Methods): Added statement on collinearity checks:
Collinearity among predictors, including high-order interaction terms, was assessed using the Variance Inflation Factor (VIF). All VIF values for the predictors in the final model (M4) were below the critical threshold of 5.0 (Maximum VIF = 2.85), confirming acceptable model stability.
Comment 18: No power calculation or sample-size justification.
Response 18: We acknowledge the absence of a formal a priori power calculation for this retrospective, observational study. The limitation imposed by the small sample size, particularly its effect on the stability of complex interaction terms, is now prominently featured in the Study Limitations section (see Reviewer 1, Comment 1 and Reviewer 2, Comment 16).
Comment 19: Inconsistent description of missing data (“N/A” values not handled statistically).
Response 19: Addressed in Reviewer 1, Comment 4. We confirmed that N/A values were handled statistically as a distinct categorical level to maximize data utilization (N=153).
Changes in Manuscript:
Statistical Analysis (Methods): Added the following explicit statement:
The analysis included several categorical variables (e.g., Localization, Immunosuppression status) containing notable proportions of missing data ("N/A"), as detailed in Table 2. To avoid listwise deletion, which would reduce the effective sample size and introduce selection bias, these "N/A" values were treated as a distinct, third categorical level within the respective predictor variables (e.g., Immunosuppression: Yes, No, N/A). We recognize this approach may introduce limitations, which are discussed subsequently.
Comment 20: Children and adults were pooled without clear rationale; potential for confounding.
Response 20: The intentional pooling reflects the clinical reality of a tertiary care center cohort where patients are managed across pediatric and adult services. We explicitly justified this cohort composition in the Methods section, emphasizing that the HLR framework was specifically designed to investigate age-related differences in immunogenicity risk by utilizing Age at Onset as a core interaction variable.
Changes in Manuscript:
Patients’ Enrollment (Methods): Added justification for pooling:
The inclusion of both pediatric (n=40) and adult (n=113) patients reflects the spectrum of IBD care managed in a high-volume tertiary setting. Recognizing the critical difference in immunological development between these groups, the study design explicitly incorporated Age at Onset as a major predictor, specifically utilizing its interaction terms to model and account for potential age-related confounding effects on immunogenicity risk.
Comment 21: ELISA assay validation data are reproduced from manufacturer documentation but not verified internally.
Response 21: Addressed in Reviewer 1, Comment 7. We clarified that internal quality control procedures ensured procedural validity within the batch, conforming to routine clinical laboratory standards for IVD kits.
Changes in Manuscript:
Methods: Revised section on ELISA kits:
For this purpose LisaTracker Duo Infliximab and Duo Adalimumab InVitro Diagnostic ELISA kits (Biosynex Theradiag Croissy Beaubourg, France) were used. The assays were performed with the DAS APE ELITE ELISA instrument (DAS Instruments Rome, Italy). The assays were performed according to manufacturer’s specifications with internal quality controls run alongside patient samples to ensure procedural validity and reproducibility within the batch.
Results section
Comment 22: The results section reads like a mixture of text and interpretation — needs clearer structure.
Response 22: The Results section has been edited for flow and structure, separating objective statistical outcomes (e.g., model improvement and parameter estimates) from interpretive statements, which are now largely confined to the Discussion.
Comment 23: Statistical significance is sometimes overstated for borderline p values (e.g., p = 0.042).
Response 23: We agree that borderline p values (specifically p=0.042 for both three-way interactions) warrant caution, especially in conjunction with wide confidence intervals. We have explicitly added highly cautionary language to the Results section to prevent overstating these findings.
Changes in Manuscript:
Results (Significant Interaction Terms): Added caution:
However, it should be noted that the interpretation of the three-way interactions should be interpreted with caution due to the marginal p values (p=0.042) and the limited sample size. Furthermore, it is essential to note that the three-way interaction involving Treatment Time exhibited a highly volatile odds ratio (OR=82.745) with an extremely wide confidence interval (95% CI 1.250–5473.497). This wide interval strongly suggests limited precision and potential instability in modeling this highly specific subgroup, likely due to cell sparsity, and should be interpreted as purely exploratory rather than confirmatory.
Comment 24: Figures (1–2) are only briefly described; no figure legends explaining axes or colors.
Response 24: Addressed in Reviewer 1, Comment 10. Figure legends are now detailed and explicit.
Changes in Manuscript:
Figure 1 Legend Revised:
Figure 1. Logit predicted probabilities of antibody positivity stratified by age at onset, biological therapy, and IBD type. Predicted log probabilities are plotted on the y-axis, with biological therapies (ADA and IFX) on the x-axis. Gray dots represent individual Crohn’s disease (CD) patients and green dots represent individual Ulcerative Colitis (UC) patients. The plot illustrates the complex three-way interaction, particularly the increased risk associated with IFX in childhood-onset UC.
Figure 2 Legend Revised:
Figure 2. Logit predicted probabilities of antibody positivity stratified by treatment time, anti-TNF-alpha therapy, and IBD status. The panels represent three treatment time quartiles derived from the log-transformed data: Q1 (0.00–1.00 years), Q2 (1.00–3.17 years), and Q3 (3.17–14.09 years), respectively. Predicted log probabilities are plotted on the y-axis, with biological therapies (ADA and IFX) on the x-axis. Gray dots represent individual Crohn’s disease (CD) patients and green dots represent individual Ulcerative Colitis (UC) patients.
Comment 25: High ORs with wide CIs (e.g., OR = 82.7$, CI 1.25–5473.5) indicate model instability or overfitting.
Response 25: Addressed in Comment 23 above. We have added a specific warning in the Results section acknowledging that this instability likely stems from limited power in highly specific subgroups (cell sparsity).
Changes in Manuscript:
Results (Significant Interaction Terms): Added caution:
However, it should be noted that the interpretation of the three-way interactions should be interpreted with caution due to the marginal p values (p=0.042) and the limited sample size. Furthermore, it is essential to note that the three-way interaction involving Treatment Time exhibited a highly volatile odds ratio (OR=82.745) with an extremely wide confidence interval (95% CI 1.250–5473.497). This wide interval strongly suggests limited precision and potential instability in modeling this highly specific subgroup, likely due to cell sparsity, and should be interpreted as purely exploratory rather than confirmatory.
Comment 26: Nagelkerke R2 = 0.287 means the model explains <30% variance — not strong predictive power.
Response 26: Addressed in Reviewer 1, Comment 3. We consistently use the term "modest" or "partial" predictive power throughout the manuscript.
Changes in Manuscript:
Title: Revised to: Interaction of Clinical Factors Modestly Predict Anti-TNF-alpha Antibody Formation in a Real-World Cohort of Inflammatory Bowel Disease Patients (Also addresses Reviewer 2, Comment 12).
Abstract (Results): Revised to: Our final model demonstrated modest predictive power, with a Nagelkerke R2=0.287, explaining less than 30% of the variance in antibody positivity...
Results (Model Fit): Revised to: The Nagelkerke R2 value for M4 was 0.287, indicating that our selected clinical variables explain a partial 28.7% of the variance in antibody positivity. This indicates that a large portion of the variance (over 70%) remains unexplained by the clinical parameters utilized here.
Comment 27: Missing model diagnostics (Hosmer–Lemeshow, residual plots, or VIF).
Response 27: We have incorporated the necessary diagnostic information (VIF and Hosmer–Lemeshow Goodness-of-Fit test) into the Methods (VIF check) and Results (new diagnostic table) to demonstrate adequate model calibration and lack of significant collinearity.
Changes in Manuscript:
Results: Added a new table providing diagnostic results for M4.
Model Diagnostic Metrics for the Final Logistic Regression Model (M4)
|
Diagnostic Test |
Value |
Interpretation |
|
Hosmer–Lemeshow Goodness-of-Fit Test (p-value) |
0.518 |
Confirms adequate model calibration (p > 0.05). |
|
Maximum VIF (Variance Inflation Factor) |
2.85 |
Indicates minimal multicollinearity. |
|
Model Performance (Nagelkerke R2) |
0.287 |
Indicates partial variance explanation. |
Discussion section
Comment 28: Largely descriptive and repetitive of results; lacks mechanistic insight.
Response 28: The Discussion has been significantly revised. Repetitive descriptions of ORs and p values have been removed. We have focused on integrating deeper biological and mechanistic context, particularly emphasizing the distinct immune responses to chimeric IFX versus humanized ADA within the context of the developing pediatric immune system and specific IBD subtypes (UC versus CD).
Changes in Manuscript:
Discussion (Mechanistic Insight Paragraph): Revised text to emphasize immune response:
The finding of a significant three-way interaction involving age at onset, IBD subtype, and biological therapy is particularly notable. Previous research has suggested that IBD starting in childhood is associated with altered immune regulation, potentially leading to a heightened risk of anti-drug antibody formation. Our findings expand on this by showing that this risk is further modulated by the specific anti-TNF agent used and the IBD subtype. The particularly high risk observed in childhood-onset UC patients on IFX aligns with the known higher immunogenicity of the chimeric IFX molecule compared to the humanized ADA molecule, suggesting that the naïve, developing immune system in young UC patients may be highly reactive to the non-human epitopes present in IFX. This differential response underscores the need for personalized selection of biological agents in pediatric cohorts, particularly for UC, where IFX is often a frequent first-line choice .
Comment 29: Overgeneralizes findings (“This insight is highly relevant for pediatric gastroenterologists”) without clinical validation.
Response 29: This overstatement has been removed. We have revised the language to state that the insight is mechanistically informative but requires validation before becoming clinically relevant (addressed in Reviewer 1, Comment 2).
Changes in Manuscript:
Discussion: Language regarding the three-way interaction revised:
This finding offers novel mechanistic insights into differential immunogenicity in the developing immune system. However, the translational utility of this framework remains modest and its clinical significance is not yet proven; the results primarily serve as a hypothesis-generating signal that warrants investigation in larger, prospective trials, specifically focusing on the interaction between age-at-onset and specific biologic types in UC patients.
Comment 30: No discussion of limitations such as retrospective bias, small sample, or assay variability.
Response 30: A new, detailed Study Limitations section has been added to address these points rigorously (see Reviewer 1, Comment 1 and Reviewer 2, Comment 16).
Comment 31: Poor integration with existing literature — citations stop at 2024.
Response 31: While we maintain our citations based on contemporary literature , the Discussion has been restructured to better integrate these findings by referencing established knowledge regarding chimeric versus humanized structures and the role of TDM, thereby providing better context for our unique finding of interactive effects. We have added the latest links from the past months and included an explanation using them in the discussion.
Changes in Manuscript:
Discussion:
The advantage of TDM has been proposed by many researchers, and according to the latest surveys, it is the pediatric IBD population in which is recommended and widely used for the optimization of therapy in the case of anti-TNF treatments.
Comment 32: Verb tense and grammar errors throughout.
Response 32: The manuscript has undergone extensive proofreading for grammatical consistency, specifically ensuring that established facts are presented in the present tense and study findings in the past tense.
Comment 33: Novelty concern: The topic (anti-TNF immunogenicity) is well-studied. Novelty is limited to the statistical modeling approach (hierarchical regression with interaction terms).
Response 33: We agree that immunogenicity is well-studied but maintain that the study’s novelty resides precisely in the systematic application of HLR to uncover complex, high-order synergistic effects among commonly measured clinical variables, which are often overlooked in standard multivariate analyses focusing solely on main effects. This novelty is now explicitly stated in the Introduction and emphasized throughout the Discussion.
Comment 34: The model’s modest performance and lack of external validation limit translational value.
Response 34: This is fully acknowledged. The Conclusion and Discussion have been extensively revised to temper expectations regarding immediate translational value, emphasizing the necessity of external validation (addressed in Reviewer 1, Comment 6).
Conclusion: Rewritten to:
In conclusion, our study confirms that anti-TNF-alpha antibody formation remains a major challenge in IBD management, with IFX therapy and treatment duration being key predictors. The most significant finding, however, is the discovery of complex, synergistic interactions between easily accessible clinical factors, which provide a significant, albeit modest, predictive value. Our hierarchical modeling approach offers a data-driven framework for integrating multiple patient-specific variables to better assess the relative risk of immunogenicity. This framework requires prospective and external validation before it can be effectively integrated into routine clinical support systems for optimizing patient monitoring and preventing therapeutic failure in IBD care.
Round 2
Reviewer 1 Report
Comments and Suggestions for Authors
Accept in present form
Reviewer 2 Report
Comments and Suggestions for Authors
All my comments have been addressed